# Recurrent Campylobacter Enteritis in Patients with Hypogammaglobulinemia: Review of the Literature

**DOI:** 10.3390/jcm9020553

**Published:** 2020-02-18

**Authors:** Iris Najjar, Florina Paluca, Konstantinos Loukidis, Philip E. Tarr

**Affiliations:** 1University Department of Medicine, Kantonsspital Baselland, Bruderholz, University of Basel, 4101 Bruderholz, Switzerland; iris.najjar@gmail.com (I.N.); florina.paluca@luks.ch (F.P.); loukidiskonstantinos@gmail.com (K.L.); 2Infectious Diseases Service, Kantonsspital Baselland, Bruderholz, University of Basel, 4101 Bruderholz, Switzerland; 3Department of Medicine, Luzerner Kantonsspital, 6004 Luzern, Switzerland; 4Hematology Service, Kantonsspital Baselland, Bruderholz, University of Basel, 4101 Bruderholz, Switzerland

**Keywords:** *Campylobacter* enteritis, campylobacteriosis, recurrent, hypogammaglobulinemia, rituximab

## Abstract

Recurrent *Campylobacter* enteritis is a well-recorded complication of primary hypogammaglobulinemia but has only rarely been reported with other types of immunodeficiency, and no cases have been reported after rituximab-associated secondary hypogammaglobulinemia. We therefore reviewed our local microbiology laboratory databases and conducted a literature search, to provide a detailed characterization of the immunodeficiency states associated with recurrent *Campylobacter* enteritis. Published cases had primary hypogammaglobulinemia, most frequently in the setting of common variable immunodeficiency, x-linked agammaglobulinemia, and Good syndrome. No cases were identified in the literature after rituximab or secondary hypogammaglobulinemia. We report a 73-year-old patient with recurrent *Campylobacter* enteritis and hypogammaglobulinemia in the setting of non-Hodgkin lymphoma, chemotherapy, and maintenance rituximab. Physicians should be aware of the association of recurrent *Campylobacter* enteritis and immunodeficiency, most commonly in primary hypogammaglobulinemia. Rituximab alone may not be sufficiently immunosuppressive for recurrent campylobacteriosis to occur; additional factors, including hematological malignancy and its treatment, appear necessary. Patients with recurrent *Campylobacter* enteritis and those starting rituximab should be investigated for hypogammaglobulinemia and B-lymphopenia.

## 1. Introduction

In healthy individuals, campylobacteriosis presents variably with diarrhea, abdominal pain, and fever. Symptoms may resolve without antimicrobial treatment, and complications such as bacteremia are uncommon. Hypogammaglobulinemia has been associated with recurrent, prolonged, and complicated campylobacteriosis. Successful treatment often requires antibiotics and intravenous immune globulin replacement (IVIG) [1]. While it is a well-recorded complication in primary hypogammaglobulinemia, recurrent campylobacteriosis in patients with secondary hypogammaglobulinemia has not previously been recorded and is rare with other types of immunodeficiency. 

Here, we provide a detailed review of the literature of recurrent *Campylobacter* enteritis in the setting of immunodeficiency. *Campylobacter* bacteremia has previously been reviewed [2] and may occur with [3,4] or without [3] clinically apparent gastroenteritis. Our review shows that recurrent *Campylobacter* gastroenteritis has been reported most commonly in the setting of primary hypogammaglobulinemia and only rarely in other immunodeficiency states. We report the case of a man who developed recurrent *Campylobacter* enteritis, in the setting of secondary hypogammaglobulinemia due to non-Hodgkin lymphoma and repeated administration of rituximab as maintenance lymphoma treatment. To our knowledge, despite the frequency of rituximab use and consequent (“secondary”) hypogammaglobulinemia, recurrent *Campylobacter* enteritis in this context has not previously been reported. Healthcare providers should be aware of the association of recurrent campylobacteriosis and immunodeficiency. Screening for hypogammaglobulinemia is now recommended prior to starting rituximab. 

## 2. Methods

In order to identify local cases, we searched our local microbiology laboratories (Kantonsspital Baselland, University Hospital Basel, Kantonsspital Luzern) for patients in whom *Campylobacter* was recovered ≥2 times over a ≥90-day period. To identify patients in the published literature, a PubMed search (no time limitation, all languages) was done. Search items included *Campylobacter*, hypogammaglobulinemia, immunocompromised, immunodeficiency, rituximab, recurrent, and relapse. Additional cases were identified through bibliographic review of retrieved publications. Recurrent *Campylobacter* gastroenteritis was defined as ≥2 episodes of clinical gastroenteritis with either positive blood or stool cultures, separated by an interval of ≥90 days, in order to account for potentially prolonged stool excretion of *Campylobacter* [4]. According to current guidelines, hypogammaglobulinemia was defined as decreased serum levels of immunoglobulin G (IgG) (≥2 standard deviations below the mean for age), in combination with a decrease of ≥1 other isotype, either immunoglobulin M (IgM) or immunoglobulin A (IgA) [5]. Cases were excluded if documentation was insufficient for review. 

## 3. Results

### 3.1. Investigations in Local Microbiology Laboratories

One case of recurrent *Campylobacter* infection was identified in the microbiology laboratory of Kantonsspital Baselland (database review 2009–2018) and one case was identified in Luzerner Kantonsspital (1 January 2017–30 June 2019). Both patients presented twice with self-limited *Campylobacter* diarrhea. The first patient was an 82-year-old male with episodes in 2009 (the subspecies was not defined) and 2011 (*C. jejuni*). Normal IgG levels were measured during the first episode, and liver cirrhosis child B was diagnosed. He was under no immunosuppressive drugs, and no recurrent diarrhea or other infections were documented. The second patient was a 69-year-old woman with two episodes of self-limited *C. jejuni* gastroenteritis, in November 2018 and March 2019. Immunoglobulin levels were not measured. She was under chronic low-dose corticosteroid therapy for inflammatory bowel disease and had no recurrent infections. No cases of recurrent *Campylobacter* enteritis were recorded in the microbiology laboratory of the University Hospital in Basel, Switzerland. 

### 3.2. Literature Review

We identified 45 cases of recurrent *Campylobacter* infection in patients with hypogammaglobulinemia in the literature. Of these, we excluded 31 cases, either because patients presented only with extraintestinal *Campylobacter* manifestations (cellulitis [6,7,8,9,10,11,12], arthritis [13], ureteric colic [8], rash [14], pericarditis [15], and spondylodiscitis [16]), or because only one episode of enteritis was documented [17,18,19], because the time interval between stool cultures was not documented or was <90 days, or because criteria for hypogammaglobulinemia were not met or not recorded [2,20,21]. Therefore, 14 cases of hypogammaglobulinemia and recurrent *Campylobacter* gastroenteritis form the basis of this review (Table 1). Of these, six patients had common variable immunodeficiency (CVID) [22,23,24,25], four had X-linked hypogammaglobulinemia (XLA) [26,27,28,29], and two had immunodeficiency with thymoma (Good syndrome) [24,30]. In two cases, the nature of hypogammaglobulinemia was not reported [31,32] but was likely primary, as no secondary causes were reported, and because thrombocytopenia and autoimmune hemolytic anemia suggested CVID in one of these patients [32]. No published cases of recurrent *Campylobacter* infection in the setting of secondary hypogammaglobulinemia were found. 

In 8 of the 14 patients, bacteremia was also recorded [25,26,27,28,29,30,31]. All patients received at least one course of antibiotics except for one case where therapy was not mentioned [31]. In seven patients, IVIG replacement therapy [24,26,28,29,30,32] was recorded before recurrent *Campylobacter* enteritis was noted; in six cases, IVIG was not mentioned [22,23,27,28,31], and in one case it was started after hypogammaglobulinemia was diagnosed because of recurrent *Campylobacter* infections [25]. In seven patients, diarrhea was resolved [24,25,26,30,32], while in six patients, including our patient, symptoms improved [22,23,28,29]. One patient continued to have fever bouts after completion of antimicrobial treatment [28]. In one case, the clinical outcome was not documented [31]. One patient died in the setting of sepsis and multiple organ failure [27].

## 4. Illustrative Case Report

A 73-year-old man was diagnosed with a follicular non-Hodgkin lymphoma in 1997. The clinical course of the patient is summarized in Figure 1. He was previously healthy, with no history of recurrent infections. He was treated with cyclophosphamide, hydroxydaunorubicin, oncovin and prednisolone (CHOP), followed by six-monthly rituximab until 1998. He relapsed in 1999, 2003, 2005, and 2012 and received chemotherapy with chlorambucil and fludarabine, followed by four cycles of rituximab (1999); fludarabine and rituximab (2003 and 2005); and bendamustin and rituximab followed by maintenance rituximab every two months for two years (2012–2014). 

Diarrhea was first recorded in 2012; it lasted intermittently for three weeks and resolved without antibiotics. Stool cultures grew *Campylobacter jejuni*. In April 2015, another episode of self-limiting diarrhea yielded *C. jejuni*. In May 2016, diarrhea of >4 weeks duration was noted; stool culture grew *C. jejuni*. Azithromycin was given for five days and diarrhea improved. Hypogammaglobulinemia was first noted at this time (undetectable IgA, IgG at 1.8 g/L), and IVIG replacement therapy (0.2 g/kg every four weeks) was started. 

In October 2016, diarrhea recurred, again without identifiable exposure. Stool culture grew *C. jejuni* resistant to azithromycin and ciprofloxacin. Treatment with intravenous ertapenem for 14 days was given, and diarrhea was resolved. In December 2016, diarrhea recurred, and *C. jejuni* grew in stool culture susceptible to azithromycin. The patient denied drinking raw milk, he had no pets, and did not use any proton pump inhibitors. He denied recurrent sinusitis, bronchitis, or other infections. Ultrasound of the abdomen was unremarkable. Computed tomography of the abdomen showed no signs of recurrent lymphoma, no hepatosplenomegaly, and no gallstones or ascites. The patient was afebrile, and no blood cultures were obtained. Following treatment with azithromycin for three days, diarrhea was temporarily resolved but recurred a week later. Stool cultures were negative, stool microscopy showed no evidence for parasites or protozoa, and a stool test for *C. difficile* toxin was negative. A multiplex PCR examination of a stool specimen was positive for *C. jejuni* and negative for other pathogens. A second stool culture one week later yielded *C. jejuni* susceptible to azithromycin. Colonoscopy was performed, and biopsies showed acute inflammation of the distal ileum, cryptitis, and crypt-abscesses of the colon without any sign of lymphoma or microscopic colitis. Treatment with azithromycin 250 mg for three days was given, and symptoms improved. 

In January 2017, in the setting of IVIG replacement (0.2 g/kg) every three weeks, undetectable immune globulin trough levels were noted. Therefore, the IVIG interval was shortened to two weeks. The serum IgG level remained below 6.5 g/L, so that the IVIG dose was increased to 0.3 g/kg every two weeks in April 2017, followed by serum IgG trough level of 9.5 g/L (August 2017). Stool cultures remained negative in April 2017. At last follow-up (August 2019), the patient has not had any further *Campylobacter* enteritis episodes and has not required any further antibiotic therapy. He describes his quality of life and stool habits as significantly improved with IVIG every two weeks. Depending on what he eats, he has between two and six daily episodes of typically semi-formed stools. 

## 5. Discussion

Our literature review of recurrent *Campylobacter* enteritis and immunodeficiency yielded 14 cases associated with hypogammaglobulinemia [22,23,24,25,26,27,28,29,30,31,32], all of which were considered to be primary, and most commonly included CVID and XLA [1]. While hypogammaglobulinemia is a well-recorded complication of rituximab treatment [33], there are no other published reports of recurrent or persistent campylobacteriosis in this context. We report on a patient with recurrent *Campylobacter* enteritis in the setting of hypogammaglobulinemia that is attributable most likely to immunodeficiency in the setting of non-Hodgkin lymphoma, previous chemotherapy, and maintenance rituximab. 

### 5.1. Recurrent Campylobacteriosis 

*Campylobacter* species are among the most common pathogens in human bacterial gastroenteritis [34]. In immunocompetent patients, campylobacteriosis may be self-limiting without antimicrobial treatment and complications such as bacteremia are uncommon. In contrast, in patients with hypogammaglobulinemia, bacteremia may occur more frequently, and antibiotics and IVIG are typically required for symptom control [2]. This underscores the importance of humoral immune mechanisms in the defense against *Campylobacter* infections.

In Canada, 1%–2% of the population may have symptomatic campylobacteriosis per year, but reported cases may represent only 2%–5% of the actual incidence [35]. In the general population, prolonged asymptomatic intestinal carriage with *Salmonella* and *Campylobacter* is well recorded, and may occur more frequently in patients with primary immunodeficiencies [36,37]. This clearly is different from recurrent symptomatic campylobacteriosis, which is rare [2]. In a retrospective analysis over 10 years, 1.2% of patients in Québec, Canada, presenting with *Campylobacter* enteritis experienced a second episode in the following five years [4]. No data on co-morbidities were available. Consistent with this low recurrence rate, we were able to identify only two patients with ≥2 stool cultures positive for *Campylobacter* species in the microbiology laboratories serving our hospitals over a >10-year time period. Both patients should be considered somewhat immunosuppressed (liver cirrhosis, low-dose steroids), but they had no recurrent infections and no hypogammaglobulinemia documented. 

*Campylobacter* recurrence most often is due to a second, independent episode of foodborne exposure, in the setting of high genetic and thus antigenic variability of *Campylobacter* species [38]. *Campylobacter*-specific antibodies may decrease within a few months after a single enteritis episode, leaving the patient with a lack of long-term immunity [39]. However, *Campylobacter* enteritis is less frequent in developing countries, although exposure to *Campylobacter* is higher [40]. In fact, not all *Campylobacter* infections are symptomatic, and it seems that immunity against symptomatic campylobacteriosis correlates with the concentration of IgA-antibodies [41]. Moreover, in Egypt, 15% of 132 examined asymptomatic children were *Campylobacter* carriers [42]. This suggests a certain degree of acquired immunity induced by multiple exposures [42]. Consistent with this notion, travelers traveling to developing countries show higher incidences of *Campylobacter* enteritis than the local population [43]. 

### 5.2. Recurrent Campylobacter Enteritis in Immunodeficient Persons

While complicated courses of *Campylobacter* infection have been associated with other forms of immunodeficiency and underlying conditions such as liver disease, solid and hematologic malignancies, chronic obstructive pulmonary disease, and heart disease [2], we found only a few cases of recurrent *Campylobacter* enteritis associated with immunodeficiency other than primary hypogammaglobulinemia. One case was associated with advanced human immunodeficiency virus (HIV) infection [44], one case with rheumatoid arthritis and methotrexate treatment [17], and one case with systemic lupus erythematodes (SLE) and treatment with low-dose cortisone [20]. In the HIV+ patient, low levels of *C. jejuni*-specific antibodies were recorded [44]. The patient with rheumatoid arthritis had low serum levels of total gamma globulin (levels of IgA, IgM, and IgG were not reported), and the patient with SLE had isolated IgA- and IgM hypogammaglobulinemia, with normal IgG serum levels noted on two occasions. These patients did not fulfill international criteria for hypogammaglobulinemia [5], but nonetheless had evidence of impaired humoral immunity. More specifically, IgA antibodies play important roles in mucosal immunity and may be instrumental in the clearance of *C. jejuni* from the intestinal tract [20]. Oksenhendler and colleagues reported that *Campylobacter* enteritis was more frequently seen in CVID-patients with undetectable serum IgA levels compared to CVID-patients with detectable IgA levels [1]. Infants who are breastfed developed *C. jejuni* diarrhea less frequently than non-breastfed children [45], and the authors suggested partial protection of these infants via IgA antibodies of maternal origin. Our patient also had an IgA deficiency. Each of these findings underscores the notion that the humoral and mucosal immune system plays a key role in the control of *Campylobacter* infection. Finally, we are not aware of detailed analyses of immunological differences that may predispose patients to either *Campylobacter* enteritis, bacteremia, or arthritis. Indeed, gastroenteritis and bacteremia can occur in the same patient, at different times, and intestinal carriage is a risk factor for bacteremia in immunocompromised patients [36,46].

### 5.3. Infections Following Rituximab 

Many thousands of patients with rheumatoid arthritis (RA) have received rituximab in the past 15 years [47], but no recurrent campylobacteriosis cases have been reported. Moreover, the reactivation of hepatitis B after rituximab is essentially limited to oncology patients [48,49,50]. This suggests that rituximab alone may not be sufficiently immunosuppressive in most patients for recurrent campylobacteriosis or hepatitis B reactivation to occur [51]: immunosuppressive cancers (i.e., non-Hodgkin lymphoma) and concomitant or prior chemotherapy seem to be necessary co-factors for these complications to occur. Our patient received rituximab repeatedly over a period of 17 years (1997–2014) for the treatment of non-Hodgkin lymphoma, but previously had received different immunosuppressive chemotherapeutic agents, including fludarabine and cyclophosphamide that likely contributed to persistent immunodeficiency. An additional factor may be the higher individual and cumulative dose of rituximab that tends to be used in lymphoma therapy compared to the treatment of RA [52,53,54,55].

Rituximab, a monoclonal anti-CD20-antibody, has been increasingly used in a variety of conditions, including autoimmune disorders and hematologic malignancies. Rituximab binds to CD20 on the surface of B-cells, inducing antibody-dependent and cell-mediated B-cell depletion. Rituximab is typically well tolerated, but occasional patients with opportunistic and non-opportunistic infections [56,57] have been reported, including pneumocystosis, invasive aspergillosis, invasive candidiasis, cutaneous herpes zoster, cytomegalovirus pneumonia and esophagitis, and progressive multifocal leukoencephalopathy [56]. 

We identified seven previously published cases of *Campylobacter* infection in the context or rituximab treatment [58,59,60,61]. Six of these cases were due to *Campylobacter fetus* [58,59,61], a relatively rare subspecies associated with bacteremia, typically in immunocompromised patients. Hypogammaglobulinemia was documented in five of these seven patients, and symptoms ranged from septic arthritis to cellulitis, acute cholecystitis, and sepsis. However, *Campylobacter* recurrence was recorded in none of these patients. 

We found one additional case of recurrent *Campylobacter* infections in a patient with autoimmune hemolytic anemia (AIHA) who received corticosteroids and later rituximab for a period of six years [25]. Immunodeficiency was suspected because of recurrent *Campylobacter* infections, and hypogammaglobulinemia was then diagnosed. 

### 5.4. Infections and Hypogammaglobulinemia Following Rituximab 

The combination of rituximab with other chemotherapeutic drugs and its repeated application, most notably in the setting of maintenance treatment to reduce the risk of lymphoma recurrence, as in our patient, may increase the incidence of infection [52,62]. Hypogammaglobulinemia is a known complication of B-cell depletion by rituximab. The reported incidence varies widely in the literature and depends on the underlying disease [33,52,63,64]. Overall, persistently low IgG levels are rare after rituximab (approximately 1% of patients). A preliminary report suggesting that hypogammaglobulinemia after rituximab may be genetically determined needs independent confirmation [65]. Patients with malignant disease may have a higher risk of post-rituximab hypogammaglobulinemia than patients with non-malignant disease [54]. In a retrospective study, Casulo et al. [52] identified 211 patients with B-cell lymphoma who had immunoglobulin levels measured before and after rituximab treatment. A total of 39% developed hypogammaglobulinemia, and 7% developed symptomatic hypogammaglobulinemia, defined as ≥2 non-neutropenic infectious episodes within six months after rituximab. Symptomatic hypogammaglobulinemia occurred more often in patients with repeated courses of rituximab, maintenance therapy, and in combination with purine analogues [62,64,66], as it was the case in our patient. Reported non-opportunistic infections after rituximab and hypogammaglobulinemia mostly involved the respiratory tract and sinuses [33,64]. Opportunistic infections did not occur more frequently in patients with hypogammaglobulinemia compared to those without [56]. The authors estimated the overall risk of infection to be threefold higher in patients with persistent hypogammaglobulinemia than in patients without [62]. 

In addition to *persistent* hypogammaglobulinemia, *delayed* hypogammaglobulinemia has also been reported to occur, several years after the last dose of rituximab [59,67]. The data suggest a progressively increasing risk of low IgM, IgG, and/or IgA levels with cumulative cycles of rituximab [68]. Additional factors, such as older age and corticosteroids, may be present [68]. 

Moreover, patients may have low IG levels and recurrent infections prior to rituximab treatment [69,70,71,72]. These patients may be more likely to develop persistent hypogammaglobulinemia after treatment, suggesting preexisting immune dysfunction. CVID is associated with an increased frequency of lymphoma, which sometimes occurs prior to the diagnosis of CVID [1,73], making it difficult to differentiate between rituximab- or lymphoma-associated hypogammaglobulinemia and CVID. In addition, patients with hypogammaglobulinemia following rituximab may have immune dysfunction that preexisted the rituximab administration, including CVID [72]; CVID is unlikely in our patient because he had no history of recurrent infections (therefore, IG levels prior to rituximab treatment were not measured, as in most cases of rituximab-associated hypogammaglobulinemia [72]). 

### 5.5. IVIG Replacement for Rituximab-Associated Hypogammaglobulinemia 

In primary hypogammaglobulinemia, IVIG typically leads to a decrease in infectious complications and an increase in life expectancy [1]. Barmettler et al. recently recorded a decrease in infections in patients with rituximab-associated hypogammaglobulinemia with increasing IVIG doses [72]. In our patient, diarrhea improved significantly and subjective quality of life increased considerably with IVIG replacement at higher dosage and shorter intervals, and *Campylobacter* was successfully eradicated from the patient’s stool cultures. The need to increase the IVIG dosage might be explained by the low concentration of IgA contained in IVIG preparations, which seems to be central for the control of *Campylobacter* infection. 

## 6. Conclusions

Physicians should be aware of the association of recurrent campylobacteriosis and immunodeficiency, especially humoral immunodeficiency. Patients with recurrent enteritis (or with a first episode of *Campylobacter* bacteremia) should be evaluated for humoral immunodeficiency by measuring serum immune globulin levels and circulating B-cells [2]. This should routinely be done also before administering rituximab, in order to identify patients with undiagnosed preexisting hypogammaglobulinemia, typically those with chronic lymphatic leukemia or lymphoma, and occasionally patients with undiagnosed CVID. In these patients, “subclinical immunodeficiency” might be unmasked by rituximab, and they may be at an increased risk of infection after rituximab is given [51,72]. In patients who develop recurrent infections after rituximab, new or worsened hypogammaglobulinemia should again be looked for, and specialist referral and IVIG replacement therapy should be considered. 

## Figures and Tables

**Figure 1 jcm-09-00553-f001:**
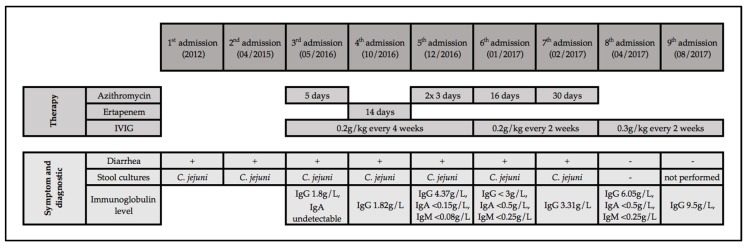
Clinical course of our patient with recurrent *Campylobacter* enteritis. Notes: IgA, immunoglobulin A; IgM, immunoglobulin M; IgG, immunoglobulin G; IVIG, intravenous immune globulin replacement.

**Table 1 jcm-09-00553-t001:** Characteristics of published patients with hypogammaglobulinemia and recurrent *Campylobacter* gastroenteritis.

Ref.	Age/Sex (yrs.)	Immuno-Deficiency	Clinical Features	Campylobacter Species *	Bacteremia (Number of Episodes)	Intravenous Immune Globulin Treatment (IVIG)	Outcome
[22]	42 M	CVID	Weight loss Diarrhea	*C. jejuni* (2 separate isolates, one in small bowel)	-	-	Diarrhea improved and repeat stool cultures remained negative 14 months after discontinuing antibiotics
[23]	63 M	CVID	Diarrhea	*C. jejuni* (3 separate isolates)	-	-	Diarrhea improved: stool cultures remained negative after discontinuing antibiotics
[23]	64 W	CVID	Diarrhea	*C. jejuni* (2 separate isolates)	-	-	Diarrhea improved
[24]	64 W	CVID	Diarrhea, hypo-volemic shock	*C. jejuni* (1 isolate), *C. coli* (1 isolate)	-	IVIG every 21 days	Diarrhea resolved and stool cultures remained negative 12 months after discontinuing antibiotics
[27]	39 W	CVID	Cellulitis, nausea, vomiting, rash, diarrhea	*C. jejuni* (2 separate isolates)	2	-	Symptoms resolved
[24]	83 W	Good Syndrome	Diarrhea	*Campylobacter sp* (1 isolate), *C. coli* (1 isolate)	-	IVIG every 14 days	Diarrhea resolved and stool cultures remained negative 2 and 6 months after discontinuing antibiotics
[26]	15 M	XLA	Loss of appetite, weight loss, fever	*C. jejuni* (2 separate isolates, one in gastric Antrum)	1	IVIG every 28 days	Weight gain, stool cultures remained negative 6 month after discontinuing antibiotics
[27]	24 M	XLA	Fever, nausea, cramping, vomiting	*C. jejuni* (2 separate isolates)	2	-	Died of Sepsis complicated by DIC and multiple organ failure
[28]	24 M	XLA	Fever, Diarrhea	*C. jejuni* (4 separate isolates)	4	-	Fever bouts recurred after second course of antimicrobial treatment
[30]	54 M	Good Syndrome	Diarrhea	*C. jejuni* (2 separate isolates)	2	IVIG every 28 days	Diarrhea resolved
[29]	18 M	Probable XLA	Fever, diarrhea and cellulitis	*C. jejuni* (4 separate isolates)	4	IVIG every 21 days	Diarrhea improved
[32]	34 W	Hypogamma-globulinemia, probable CVID	Diarrhea	*C. jejuni* (2 separate isolates)	-	IVIG every 14 days	Diarrhea resolved and follow up stool cultures were negative
[31]	M	Hypogamma-globulinemia, probably primary	Diarrhea, sepsis	*C. jejuni* (4 separate isolates)	4	-	not reported
[25]	30 F	Probable CVID, corticosteroids and rituximab for autoimmune hemolytic anemia	Fever, diarrhea, dyspnea, myalgia, arthralgia	*C. spp* (5 separate isolates in blood and stool cultures)	4	-	Diarrhea resolved, no recurrence during five years follow-up
Present case	73 M	Secondary hypogamma-globulinemia	Diarrhea	*C. jejuni (6* separate isolates)	-	IVIG every 21 days Then IVIG 14 days	Persistent mild diarrhea, stool cultures remained negative 6 months after discontinuing antibiotics

**Notes**: ***** The site of *Campylobacter* isolation was in stool culture unless specified otherwise. CVID, common variable immunodeficiency; DIC, disseminated intravasal coagulopathy; NA, not available; XLA, X-linked hypogammaglobulinemia.

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
