# Peer review of "Recurrent Campylobacter Enteritis in Patients with Hypogammaglobulinemia: Review of the Literature"

_jcm, 2020, doi:10.3390/jcm9020553_

Round 1
Reviewer 1 Report
Congratulations on a very well written, thorough review.
Don't you think the differences between rituximab-associated infections in lymphoma and RA (e.g, in hepatitis B reactivation) may be related to the dose used?
You choose to focus on recurrent campylobacter enteritis. Is this a different condition, immunologically, than invasive campylobacter disease (eg, arthritis and bacteremia)
Author Response
Response to Reviewer 1 Comments
1) Congratulations on a very well written, thorough review. Don't you think the differences between rituximab-associated infections in lymphoma and RA (e.g, in hepatitis B reactivation) may be related to the dose used?
Answer: This is a valid point and we have added a sentence in this regard to the revised manuscript (page 8, line 342-344).
2) You choose to focus on recurrent campylobacter enteritis. Is this a different condition, immunologically, than invasive campylobacter disease (eg, arthritis and bacteremia)?
Answer: This is an important question and we have added respective sentences to the introduction (page 2, line 57-58) and the discussion (page 8, line 327-331). Indeed, 8 of the 14 patients we report on also had bacteremia. We chose to focus on Campylobacter enteritis because of the clinical findings of our patient and because there was no other review on this topic available in the literature.
Reviewer 2 Report
Thank you for giving me an opportunity for reviewing your article. I have gone through the manuscript with an interest. Overall, the report is well written in detail. I have provided some comments below.
Comments
The major problem of this article is that, although the illustrative case and author’s interest are on the “campylobacter infection in secondary hypogammaglobulinemia”, they summarized the literature review for primary hypogammaglobulinemia. I feel an inconsistency in this point. The manuscript was submitted as review article, but it seems rather a “Case report with literature review”. This reviewer suggests a change of submission category from review article to Case Report section. (It might depend on an editor…) The authors described that they conducted a comprehensive case review in PUBMED. However, it seems not a perfect one. Please again refer to the research engine for possible articles that fulfill the definitions. Clinical course of the illustrative case is very complicated. Please provide a Figure depicting the patient’s clinical course. How did the authors differentiate agammaglobulinemia and hyperglobulinemia in this report? In some parts of the manuscript, they did not distinguish these similar, but different conditions. Please clarify this point.Author Response
Response to Reviewer 2 Comments
1) The major problem of this article is that, although the illustrative case and author’s interest are on the “campylobacter infection in secondary hypogammaglobulinemia”, they summarized the literature review for primary hypogammaglobulinemia. I feel an inconsistency in this point.
Answer: We broadly searched the literature for all immunodeficiency states that have been associated with recurrent campylobacter enteritis. Our literature review showed that primary immunodeficiency is by far the most underlying common immunodeficiency state. Consistent with this notion, these are the reports that we summarize in Table 1 and on which we base the discussion. This is our best assessment of the evidence.
2) The manuscript was submitted as review article, but it seems rather a “Case report with literature review”. This reviewer suggests a change of submission category from review article to Case Report section. (It might depend on an editor…)
Answer: The submission category “review article” seems most appropriate, because we provide far more extensive literature review and discussion than any case report would include.
3) The authors described that they conducted a comprehensive case review in PUBMED. However, it seems not a perfect one. Please again refer to the research engine for possible articles that fulfill the definitions.
Answer: We thank the reviewer for this comment. We searched PubMed again and identified 3 additional cases, 1 of which fulfilled our inclusion criteria. This case has been added to Table 1 (Gharamti et al, reference number 27) and is discussed in more detail on page 8 (line 358-361).
4) Clinical course of the illustrative case is very complicated. Please provide a Figure depicting the patient’s clinical course.
Answer: A figure 1 depicting the patient’s course has been added to the revised manuscript.
5) How did the authors differentiate agammaglobulinemia and hyperglobulinemia in this report? In some parts of the manuscript, they did not distinguish these similar, but different conditions. Please clarify this point.
Answer: We are not sure that we understand what the reviewer is referring to. Our manuscript deals with hypogammaglobulinemia and not hypergammaglobulinemia which is unrelated.
Reviewer 3 Report
I found the article very interesting, as it collects and analyzes a series of patients with recurrent Campylobacter enteritis in patients with hypogammaglobulinemia. To our knowledge, is the first work showing the current epidemiology of hypogammaglobulinemia and its associated conditions in patients with Campylobacter enteritis. It is clearly written.
While most articles of its kind present local case reports in the first place and literature review afterwards, in the present work the case report is presented after literature review, but I think this is not a problem. Also, cases collected through local laboratories do not contribute to the cohort since they do not have hypogammaglobulinemia, but this is correctly explained in the text and may help to illustrate the low prevalence of the condition.
I suggest some changes:
-Microorganisms must be written in italic and capitalized (Campylobacter jejuni, Campylobacter spp.).
-Please revise in-text abbreviations (eg IGIV appears for the first time in line 93 but the abbreviation is presented in line 118 and appears some times more in the text wihtout abbreviation. It occurs similarly with IG).
-In line 154 the sub-title 5.1 Recurrent Campylobacteriosis is not correctly formatted (it should be italic instead of bold - as in 5.1 Recurrent Campylobacteriosis)
Author Response
Response to Reviewer 3 Comments
I found the article very interesting, as it collects and analyzes a series of patients with recurrent Campylobacter enteritis in patients with hypogammaglobulinemia. To our knowledge, is the first work showing the current epidemiology of hypogammaglobulinemia and its associated conditions in patients with Campylobacter enteritis. It is clearly written. While most articles of its kind present local case reports in the first place and literature review afterwards, in the present work the case report is presented after literature review, but I think this is not a problem. Also, cases collected through local laboratories do not contribute to the cohort since they do not have hypogammaglobulinemia, but this is correctly explained in the text and may help to illustrate the low prevalence of the condition.
I suggest some changes:
-Microorganisms must be written in italic and capitalized (Campylobacter jejuni, Campylobacter spp.).
Answer: Done.
-Please revise in-text abbreviations (eg IGIV appears for the first time in line 93 but the abbreviation is presented in line 118 and appears some times more in the text wihtout abbreviation. It occurs similarly with IG).
Answer: Done.
-In line 154 the sub-title 5.1 Recurrent Campylobacteriosis is not correctly formatted (it should be italic instead of bold - as in 5.1 Recurrent Campylobacteriosis)
Answer: Done.
Round 2
Reviewer 2 Report
The revised manuscript is almost well written.